# Musashi–1—A Stemness RBP for Cancer Therapy?

**DOI:** 10.3390/biology10050407

**Published:** 2021-05-05

**Authors:** Nadine Bley, Ali Hmedat, Simon Müller, Robin Rolnik, Alexander Rausch, Marcell Lederer, Stefan Hüttelmaier

**Affiliations:** 1Department for Molecular Cell Biology, Institute for Molecular Medicine, Martin Luther University Halle/Wittenberg, Charles Tanford Protein Center, Kurt–Mothes–Str. 3A, 06120 Halle, Germany; ali.hmedat@uk-halle.de (A.H.); simon.mueller@medizin.uni-halle.de (S.M.); robin.rolnik@student.uni-halle.de (R.R.); alexander.rausch@medizin.uni-halle.de (A.R.); marcell.lederer@medizin.uni-halle.de (M.L.); stefan.huettelmaier@medizin.uni-halle.de (S.H.); 2Core Facility Imaging, Institute for Molecular Medicine, Martin Luther University Halle-Wittenberg, Charles Tanford Protein Center, Kurt–Mothes–Str. 3A, 06120 Halle, Germany

**Keywords:** RNA–binding protein, musashi–1, MSI1, stemness, cancer, cell cycle, signaling

## Abstract

**Simple Summary:**

Musashi–1 (MSI1) is an RNA–binding protein that promotes stemness properties. It was initially discovered as a regulator of neuronal development and oocyte maturation in flies and frogs. Due to its specific expression pattern with high levels during development and in a variety of cancers, MSI1 evolved as an interesting target for cancer therapy. In cancer cells, the protein mainly promotes an undifferentiated state enhancing cancer growth and therapy resistance. In this review, we summarize previous findings from development of other organisms, outline MSI1′s expression and function in different cancer entities and highlight the development of MSI1–directed inhibitors.

**Abstract:**

The RNA–binding protein Musashi–1 (MSI1) promotes stemness during development and cancer. By controlling target mRNA turnover and translation, MSI1 is implicated in the regulation of cancer hallmarks such as cell cycle or Notch signaling. Thereby, the protein enhanced cancer growth and therapy resistance to standard regimes. Due to its specific expression pattern and diverse functions, MSI1 represents an interesting target for cancer therapy in the future. In this review we summarize previous findings on MSI1′s implications in developmental processes of other organisms. We revisit MSI1′s expression in a set of solid cancers, describe mechanistic details and implications in MSI1 associated cancer hallmark pathways and highlight current research in drug development identifying the first MSI1–directed inhibitors with anti–tumor activity.

## 1. Introduction

RNA–binding proteins (RBPs) control all aspects of post–transcriptional gene expression, including RNA splicing and editing, transport and localization, mRNA turnover and translation as well as miRNA biogenesis [1,2]. Therefore, they play essential roles during development, but also serve essential roles in tumor biology by modulating essentially all hallmarks of cancer [2,3]. De–regulation of RBPs is associated with a variety of human malignancies, including solid cancers, and targeting their expression or functions provides alternative strategies for cancer therapy [4,5,6].

One of the RBPs with solid diagnostic and therapeutic potential is Musashi–1 (MSI1). MSI1 is strongly associated with stemness properties of cancer cells and de–regulated in a variety of solid cancers including gliomas. Here, we discuss functions and mechanisms directed by MSI1 in stemness and signaling, highlight its disturbed expression as well as prognostic value in solid cancers and summarize recent literature on promising targeting approaches with prospects in cancer therapy.

## 2. Musashi–1—A Conserved Stemness RBP with Unique Functions

### 2.1. Structure, Interactions and Function of MSI1 in Guiding RNA Fate

Musashi–1 (MSI1) is a member of the Musashi RNA–binding protein family consisting of two orthologs in humans and mice. Its strong expression pattern in the nervous system [7] and its primary protein structure is conserved among species including nematodes (*C. elegans*) [8], flies (*Drosophila*) [9] and vertebrates [10,11], with the human and mouse proteins sharing sequence identity on the protein level. MSI1 comprises two RNA recognition motifs (RRMs) in the N–terminal region, each containing a nuclear localization signal (NLS, Figure 1) [12,13]. RNA–binding studies revealed preferentially association to poly(G) and moderate binding to poly(U) [13]. The majority of target mRNA–binding is facilitated by RRM1 harboring preference for the consensus motif G/AU_1–3_AGU [13,14,15,16]. RRM2 was proposed to mainly stabilize protein–RNA complexes by associating to UAG [17]. The consensus motif(s) are suggested to mainly occur in single stranded, partially bulged hairpin structures, located preferentially within the 3′UTR (3′ untranslated region) of target transcripts [16].

The proper function in guiding RNA fate involves a variety of protein interactions proposed for MSI1. In embryonic stem cells (ESCs), MSI1 was reported to interact with LIN28B via its C–terminal region, resulting in elevated nuclear localization of LIN28B [18]. This was proposed to promote inhibition of let–7 miRNA maturation by LIN28B and foster self–renewal as well as pluripotency in ESCs. In HEK293T cells and *Xenopus* oocytes, MSI1 was shown to associate with both (Figure 1), the poly(A) binding protein PABP and the cytoplasmic poly(A) polymerases GLD–2 (TENT2 in human), via a domain C–terminal of RRM2 [19,20]. In the human cell models, MSI1 was proposed to compete with eIF4G–binding of PABP, resulting in impaired translation of target mRNAs in translation competent cell extracts [19]. In addition, MSI1 was proposed to recruit GLD2 to target mRNAs in *Xenopus* oocytes, resulting in elevated cytoplasmic poly–adenylation to induce and foster protein synthesis as well as mRNA stability (Figure 1) [20]. This dual role in translational repression or activation largely relies on cellular context and external cues such as progesterone promoting GLD–2 association [21]. Prospectively, studies on MSI1′s role in translational regulation in cancer cells needs to consider this dual function and external guidance cues, requiring further in–depth investigation.

Next to the modulation of mRNA translation, MSI1 has also been implicated in the regulation of mRNA turnover [22,23]. Most recently, the protein was demonstrated to impair CD44 mRNA turnover in a 3′UTR–dependent manner by limiting miRNA–mediated decay [23]. Under stress conditions such as hypoxia or platinum–based therapy, MSI1 was shown to re–localize from the nucleus to the cytoplasm, suggested to promote its oncogenic function(s) [22]. In the cytoplasm, MSI1 was proposed to either stabilize or destabilize target mRNAs in complex with AGO2 depending on complex binding in the coding region or 3′UTR, respectively [22] (Figure 1). This regulation was proposed for MSI1–directed degradation of CDKN1A (p21/WAF) and TP53 as well as stabilization of CDK4 and CCND1 (cyclin D1) transcripts. Notably, this guidance cue dependent functional plasticity supports the aforementioned observations in translational control [20,21]. Prospectively, MSI1–directed post–transcriptional control of gene expression could become even more complex involving RNA modifications, such as m6A (N6–methyladenosine), becoming more critical for cancer development [24]. In this view, it was previously shown that MSI1 promotes cancer stem cell properties of glioblastoma cells via upregulation of the m6A reader YTHDF1 (YTH N6–methyladenosine RNA–binding protein 1) [25].

MSI1 is a dual compartment protein, localized to the nucleus and cytoplasm and shuttling was reported under stress conditions [22]. Consistent with its ample, proposed roles in guiding (m)RNA fate, MSI1 is recruited to stress granules (SGs), like many RBPs [19,26,27,28]. Surprisingly, however, the RBDs are dispensable for SG–localization of MSI1, suggesting the N– and C–terminal regions to promote SG–recruitment of MSI1 [26,28]. If MSI1 affects SG formation and if SG–recruitment is linked to its roles in controlling cytoplasmic mRNA fate, however, remains largely unknown and requires further investigations.

### 2.2. Role in Development

The RNA binding protein Musashi–1 (MSI1) was originally discovered as a key player in asymmetric cell division, stem cell function and cell fate determination in *Drosophila* [7]. Already in 1994, Namamura and colleagues discovered in *Drosophila* that dMsi was essential for the development of the adult sensory organ in *Drosophila* and has been shown to be a prerequisite for the asymmetric division of sensory organ precursor cells (SOPs) [9]. While in *dMsi* wild type animals, two second order precursors were developed (a neural and a non–neuronal), the neuronal lineage was missing in *dMsi* mutants. Due to the increase in the non–neuronal lineage, more socket and/or shaft cells gave rise to the typical “double bristle” phenotype comparable to two Samurai swords that were eponymous for the protein [9]. Mechanistically, this phenotype was explained by the translational repression of the *tramtrack* mRNA by *dMsi* [15,29]. *Tramtrack* encodes for a BTB–ZF transcriptional repressor essential for photoreceptor development, repressing the neuronal lineage in the *Drosophila* eye and enteroendocrine cell specification in *Drosophila* intestinal stem cell lineages [30,31]. Later on, *tramtrack* was identified as human PLZF transcription factor (ZBTB16) [32]. While *tramtrack* mRNA was present in both SOPs, the protein was only found in the non–neuronal lineage in *dMsi* wild type animals, suggesting a translational repression for which an interaction to pRB was proposed [15,33].

The binding motif GU_3–6_G/AG of dMsi was determined by SELEX (Systematic Evolution of Ligands by EXponential Enrichment) [15]. SELEX is a screening technique selecting specific targets from a large combinatorial pool of RNA or DNA oligonucleotides by several reiterative rounds of selection and amplification [34]. The method can be applied to several RNA/DNA binding molecules including proteins and peptides, drugs, small molecules or even metal ions [35]. The motif thus obtained and its function in translational control was validated by luciferase reporter assays [15]. However, it remains still unsolved why *dMsi* represses *tramtrack* translation in the neuronal but not non–neuronal lineage, since the *dMsi* protein is present in both precursor types. Post–translational modifications and a context dependent switch of functions were thereby considered [21,36].

In *Xenopus*, MSI1 was shown to promote translation of the serine/threonine kinase and proto–oncogene MOS [37], an essential activator of the MAPK activator MEK (MAP2K1, mitogen–activated protein kinase kinase 1) in frogs and higher vertebrates [38]. Thereby, MSI1 impacts the strict temporal order of maternal mRNA translation required for meiotic cell cycle progression in oocytes [37]. The translational activation of MOS relies on MSI1 association with the polyadenylation response element in the MOS 3′UTR containing the consensus motif G/AU_1–3_AGU [14,37]. This was subsequently shown to recruit the poly(A) polymerase GLD–2 to promote polyadenylation, which in turn activates translation [20]. Furthermore, these studies provided initial evidence that MSI1′s function can be modulated on the post–translational level by external cues, resulting in a context dependent switch of function from a translational repressor to an activator or mRNA stabilizer [22,23,37].

In the mammalian system, MSI1 was identified as a marker for neuronal stem cells (NCSs) together with RBPs of the ELAVL family [39]. Subsequently, it was shown that these proteins are not only co–expressed but functionally connected, since ELAVL1 (HuR) promoted MSI1 expression [40]. A consensus binding motif G/A)U_n_AGU for MSI1 in the mammalian system, identified by SELEX, revealed high similarities to the *Drosophila* motif [14,15]. This motif is present in the 3′UTR of murine NUMB mRNA. Consistent with co–expression of MSI1 and NUMB in NSCs, it was demonstrated by a set of techniques that MSI1 regulates NUMB mRNA translation [14]. By repressing NUMB, MSI1 was suggested to promote Notch signaling in stem cells essential for the maintenance but not generation of neural stem cells [9,16,41]. This suggested a function in stem cell renewal which was subsequently validated by loss– and gain–of–function assays in cancer cells (e.g., [41,42,43,44,45]). Notably, the consensus motif identified by Imai and colleagues in the NUMB mRNA 3′UTR is utilized to identify MSI1 directed small molecule RNA–binding inhibitors such as gossypol [46].

In HEK293 cells, MSI1 was shown to modulate cell cycle progression, specifically G2/M transition, by inhibiting CDKN1A (p21/WAF) expression [47]. As for NUMB, translational repression via association of MSI1 to the CDKN1A 3′UTR was proposed. However, CDKN1A mRNA levels changed upon perturbing MSI1 abundance [47], suggesting contribution of secondary regulation or control of mRNA turnover. In support of this notion, up–regulation of both CDKN1A and NUMB upon MSI1 depletion was observed in bladder carcinoma cells [27]. Likewise, CDKN1A up–regulation upon MSI1 depletion in P19 mouse embryonal carcinoma cells promoted neuronal differentiation, rescued by additional CDKN1A depletion [47]. Despite controversy on molecular mechanisms, these findings are concise with a role of MSI1 in promoting a stem cell–like, pluripotent state in neural cells and a role in neuronal development [11,17,36].

## 3. Expression in Human Cancers

### 3.1. Expression in Human Tissue

According to its function as stemness modulator and in contrast to its homolog MSI2 that shows a more ubiquitous and persistent expression [48], MSI1 is highly abundant during murine (not shown) and human embryonal development (Figure 2A, brain; from https://www.brainspan.org/, accessed on 30 March 2021) in various organs and expression declines towards birth. In accord with its expression in *Drosophila* and *Xenopus*, MSI1 is highly abundant in stemness niches in the CNS (central nervous system) and reproductive tissue of the adult human body. MSI1 mRNA and protein is found in CNS progenitor cells, including neural stem cells [49]. Co–expression with neuronal and astrocyte intermediate filament proteins Nestin and GFAP suggest MSI1 expression in neuronal and astrocyte progenitor cells, but the protein was also reported in GFAP negative glia cells [49,50]. Notably, MSI1 was only observed in multipotent neural precursor cells, but not in newly generated postmitotic neurons, supporting its proposed roles in maintaining stem/progenitor cell properties [51]. Within fetal and adult rat testis, MSI1 is expressed in Sertoli cells supporting germ cell development [52]. In fetal and adult rat ovaries, Msi1 was detected in granulosa cells and their precursors promoting oocyte maturation and hormone production [52]. Surprisingly, Msi1 is expressed in both proliferating and nonproliferating Sertoli and granulosa cells, suggesting additional, post–mitotic roles of MSI1 in these cell types.

This specific expression pattern together with its de–regulation and functional role in a set of human cancers puts MSI1 in row with other *bona fide* oncofetal RBPs such as IGF2BP1 and 3, LIN28B or MEX proteins, representing diagnostic markers with therapeutic potential for cancer treatment [4,53,54,55,56].

### 3.2. Expression and Prognostic Value in Solid Human Cancers

In contrast to MSI2, for which oncogenic potential and expression was predominantly reported in leukemia, MSI1 expression was shown in a variety of human cancers, primarily solid cancers [57]. The reinvestigation of MSI1/2 expression in 18 human cancer transcriptomes (TCGA, The Cancer Genome Atlas; https://cancergenome.nih.gov/, accessed on 30 March 2021) and corresponding normal tissue (TCGA and GTEx; https://gtexportal.org, accessed on 30 March 2021) via the Gepia2 database [58,59] revealed selective expression of MSI1 in some cancer tissues (Figure 2B). The most prominent and significant upregulation of MSI1 is observed in low–grade glioma (LGG) and glioblastoma (GBM), and ovarian (OV) and endometrial cancers of the uterine corpus (UCEC) (Figure 2B,C). Despite variable significance of survival analyses on mRNA basis by KM plotter [60], a strong association of elevated MSI1 protein expression and adverse patient outcome was reported for all these cancers [50,51,61,62,63,64]. Mild yet not significant up–regulation of MSI1 mRNA is also seen in bulk RNA–Seq data of prostate (PRAD), esophageal (ESCA), liver (LIHC) and bladder (BLCA) cancer (Figure 2B). In support of protein–centered studies in these cancer entities [65,66,67,68], elevated expression of MSI1 is associated with an unfavorable prognosis in these malignancies, as well.

In all other cancer entities investigated, the MSI1 mRNA is barely expressed and/or remains unchanged in tumor vs. normal tissue (Figure 2B). Notably, in these cancers, MSI1 expression shows no striking association with patient survival, suggesting that elevated MSI1 expression is a conserved predictor of poor patient outcome in indicated cancers. In contrast, previous findings suggested strong MSI1 expression with prognostic value in some of these entities, including breast (BRCA), colorectal (COAD), lung (LUAD) or cervical (CESC) cancers [69,70,71,72,73,74,75]. This discrepancy between bulk RNA–Seq and IHC data could result from a minor presence of MSI1^+^ cancer stem cells (CSCs) in these tumors or a potential cross–reactivity of the MSI1–directed antibodies with MSI2 [69,72,73,74,75].

Although only partially elevated in a variety of malignancies, e.g., SKCM, MSI2 abundance is substantially higher in normal as well as tumor tissue when compared to MSI1. This is in agreement with its ubiquitous expression in human and mouse development, where MSI2 expression does not follow a strict oncofetal pattern (see Figure 2A,B). Thus, the strong oncofetal expression of MSI1, its association with reduced overall survival probability and its pro–oncogenic properties suggest MSI1 as a potent marker and promising therapeutic target in various solid cancers.

## 4. MSI1 as a Modulator of Cancer Hallmarks

The oncofetal expression of MSI1 in some cancers and its role in development as well as stem cell fate suggested MSI1 as a promising candidate target in cancer therapy. Various studies therefore aimed to evaluate its therapeutic value by exploring its role in cancer, primarily cancer–derived cell lines. To this end, some studies analyzed MSI1–associated RNAs based on RIP (RNA immunoprecipitation) and iCLIP (individual–nucleotide resolution Cross–Linking and ImmunoPrecipitation) studies [16,64,76]. iCLIP is a method to identify protein–RNA interactions using UV light to covalently cross–link protein–RNA complexes. This allows a very stringent purification with reduced background associations.

The functional impact of MSI1 in cancer was studied on the basis of loss– as well as gain–of function studies, including shRNA–, siRNA–, morpholino–directed impairment of MSI1 abundance and its overexpression [14,19,22,23,25,47,65,69,75]. Despite technical limitations and flaws of individual studies, there is an overarching agreement in that MSI1 is a potent regulator of various cancer hallmark pathways such as proliferation, apoptosis, anoikis resistance and self–renewal, migration, invasion and EMT (epithelial–mesenchymal transition), as well as tumor growth in vivo.

### 4.1. Conserved Pathways in Solid Human Cancers

Aiming to revisit MSI1′s connection to “hallmarks of cancer” gene sets, we investigated the association of MSI1 correlated genes to KEGG signaling pathways from the Kyoto Encyclopedia of Genes and Genomes in 19 TCGA RNA–Seq cohorts of solid cancers using the R2 platform (https://hgserver1.amc.nl/, accessed on 30 March 2021) (Figure 2D). This identified a strong conservation of KEGG pathways comprising genes characterized by positive association with MSI1 expression in cancer (Figure 2D). In contrast, barely any conservation was seen among genes with inverse expression to MSI1 in cancer (data not shown). For KEGG: Cell_cycle and KEGG: Basal_cell_carcinoma, we found the highest conservation with significant associations in 11 out of 18 (61%) of analyzed cancer entities and 100% or 75% conservation among the top four MSI1 de–regulated cancers LGG, GBM, OV and UCEC (Figure 2B,C). KEGG: Fanconi_anemia and KEGG: Signaling_pathways_regulating_pluripotency_of_stem_cells also showed a strongly conserved association with MSI1 in ca. 45% of all 18 tumor types and 3 out of the top four MSI1 related cancers (Figure 2B,C). KEGG: Wnt (38%), Notch (27%) and Hedgehog (17%) signaling pathways were associated with MSI1 expression; however, the conservation between cancer entities is substantially less stringent. Notably, in OV, we found a significant MSI1–correlation to genes from all top 10 KEGG pathways (Figure 2B). In GBM and LGG MSI1, association to 70% or 50% of the KEGG pathways was identified (Figure 2B).

To identify the most conserved genes related to KEGG: Cell_cycle, we compared the MSI1–correlated genes among all 11 associated cancer entities (Figure 3, red labels; please refer to Appendix A, for entire KEGG pathway map). This revealed association of MSI1 with genes throughout the entire pathway. However, most genes with significant positive association with MSI1 in cancer serve functions, primarily at G1/S transition (Figure 3, red). This is in agreement with previous findings, demonstrating a G1 arrest upon MSI1 depletion [75,77]. Interestingly, we also found CDK4 and CCND1 among the genes with highly conserved correlation to MSI1 (Figure 3) previously shown to be stabilized by MSI1 association [22].

### 4.2. MSI1, a Stemness Factor in Brain Cancers

The expression and role of MSI1 in cancer was predominantly studied in brain cancers. MSI1 was identified as marker of cancer stem cells (CSCs) arising from different brain cancers including glioma, pediatric brain cancers, medulloblastomas or astrocytomas. CSCs are thought to substantially promote cancer progression and therapy resistance to radiation or chemotherapeutics such as temozolomide [44,78,79,80,81,82]. This is associated with neurosphere formation of glioblastoma and medulloblastoma–derived cell lines and tumor growth promoted by MSI1 in vitro and in vivo [22,25,43].

In agreement, MSI1 was shown to modulate cancer hallmark pathways, including cell cycle control, Hedgehog and AKT signaling [16,43,83]. Cell cycle regulation by MSI1 was shown to involve next to CDKN1A (p21/WAF) a direct regulation of CDK4 and CCND1 [22,47]. Thereby, a novel mechanism of MSI1–directed mRNA stability control was proposed (refer to Figure 1). In medulloblastoma, MSI1 was associated with Hedgehog signaling and its depletion sensitized cancer–derived cells toward Hedgehog blockade [43]. By promoting IL6 secretion forming a self–sustaining feedforward loop with the AKT pathway, MSI1 was proposed to inhibit drug–induced apoptosis [83]. Thereby, the protein regulates both cellular signaling and cytokine secretion to create an intra– and intercellular niche for GBM to survive chemotherapy [83]. Similar mechanisms involving the de–regulation of the AKT signaling was also reported in lung cancer–derived cell lines [84].

In view of the spreading of glioblastoma cancer stem cells into surrounding brain tissue escaping cancer therapy, MSI1 was shown to promote cell adhesion, migration and invasion. This was at least partially facilitated via modulating the expression of adhesion molecules including Tensin3, ICAM1, VCAM1 and CD44 by MSI1 in glioblastoma cells [23,76,85,86]. In agreement with its conserved association with the Fanconi anemia pathway, MSI1 is also connected to DNA damage response (DDR). By promoting the hyperactivation of the DDR by increasing homologous recombination repair (HR) and evading apoptosis, MSI1 is involved in radio–resistance in vitro and in vivo [87]. This seemed to depend on the localization of the protein, since full–length but not NES/NLS–mutated MSI1 promoted cisplatin resistance in vivo [22]. In this context, the formation of stress granules is suggested to be involved in MSI1–mediated chemoresistance in refractory glioblastoma [28]. The identification of MSI1–directed inhibitors such as Luteolin [23,88] could thus represent a promising strategy to sensitize brain cancer cells for chemo– or radiotherapy.

### 4.3. A Cell Cycle Modulator in Colon Cancer

In colon cancer, MSI1 was reported as tumor marker with prognostic value [70,89]. Initially, MSI1 in the intestine was identified as a marker of crypt cells [90,91]. Along these lines, its expression was up–regulated in colonspheres [77]. In colon cancer–derived cells, MSI1 depletion suppressed the proliferation, colony formation, spheroid formation and progression of implanted colonspheres [77]. A cell cycle arrest at G0/G1 phase with upregulation of p21 expression was shown. The growth of cancer–derived xenografts from colon [41] was diminished by MSI1–directed siRNA application. This reduced tumor growth and induced apoptosis by a G2/M cell cycle arrest associated mitotic catastrophe. Changes of p21 and Notch signaling were hereby reported. Accordingly, a transgenic mouse model with forced intestinal MSI1 expression manifested a higher crypt size accompanied by enhanced proliferation [87]. Comparative transcriptomics by RNA–seq revealed the association of MSI1 with a gut stem cell signature, cell cycle, DNA replication, and drug metabolism. Therefore, CCND1, CDK6 and SOX4 were identified as MSI1 targets [92]. In cellulo, overexpression of MSI1 promoted the development of CD44+ stem cells and triggered the resistance to 5–fluorouracil [26]. Finally, the first MSI1 directed small molecule inhibitor (–)–gossypol was identified and validated in colon cancer–derived cells [46].

### 4.4. A Signaling Regulator in Female Cancers

In breast cancer cells, MSI1 promotes spheroid growth and colony formation, as well [42,69]. MSI1 was shown to drive progenitor cell expansion along the luminal and myoepithelial lineages [42]. Its expression was associated with stemness markers CD133, BMI1, SOX2, NANOG and OCT4 [69]. MSI1–directed proliferation is associated with a reduction of the secreted Wnt pathway inhibitor dickkopf–3 (DDK3) as well as increased secretion of the prolactin family member proliferin 1 (PLF1), promoting ERK activation via the IGF2 receptor (IGF2R) [42]. Due to its enhancement of ERK kinase activity, MSI1 was suggested to modulate the cross–talk between MAPK, Notch and Wnt signaling by an autocrine process that coordinates cell cycle entry. MSI1–directed cell cycle control was also reported in endometrial cancer [75]. As for other cancers, such as prostate or esophageal carcinomas [27,67], MSI1 depletion was associated with an arrest in the G1 phase accompanied by a de–regulation of CDKN1A (p21/WAF) and NUMB [75]. In breast cancer cells, however, an alternative way to activate Notch signaling by MSI1 was proposed to involve preventing the NFYA–26S proteasome axis from inactivating the NOTCH–ICD (intra cellular domain) [93]. A de–regulation of Wnt signaling was also reported for liver and cervical cancer [65,94], associated with a reduction of EMT markers such as SNAI1, SNAI2, ZEB1 or VIM in cervical cancer–derived cell lines. A link between MSI1 and drug resistance was shown in ovarian cancer–derived in which a MSI1 depletion abolished the paclitaxel resistance of A2780–derived paclitaxel resistant cells [95].

### 4.5. Control of MSI1 Expression in Cancer

The oncofetal expression and role in pluripotent stem–like cells suggests a tight control of MSI1 synthesis. However, how MSI1 expression is controlled in cancer remains largely elusive. In metastatic colorectal cancer, MSI1 expression was enhanced by Notch signaling, suggesting a feed–forward circuit, potentially involving KLF4, which was shown to bind to the MSI1 promotor [96,97]. Several studies proposed substantial regulation of MSI1 expression at the post–transcriptional level, in particular by miRNAs, including miR–125 or miR–137, both associated to tumor progression in colon cancer [84,98,99,100,101,102,103].

## 5. Targeting Musashi–1 in Cancer

Although RBPs represent ideal targets for cancer treatment due to their, in some cases, unique expression pattern, they were considered as hard to target by small molecule inhibitors due to the lack of well–defined binding pockets and the lack of catalytic activity in most RBPs. Moreover, the strong electrostatic attraction between negatively charged RNA backbone and the positively charged RBDs of RBPs, along with large interacting surface, makes it challenging to obtain small molecules inhibitors. However, drug screening and modeling approaches have overcome these obstacles, identifying a number of compounds that could prove effective in the future [6,104].

Recently, several synthetic small molecules and naturally occurring substances were identified to modulate the RNA–binding capacity of the Musashi protein family. While some of the compounds, such as Ro 08–2750 [105] or Aza–9 [106], were shown to inhibit both MSI1 and MSI2, others were more specific to MSI1.

Using a fluorescence polarization assay, (–)–gossypol was the first natural compound identified to disrupt MSI1 binding to the RNA–binding consensus motif [46]. In the past, (−)–gossypol has been extensively investigated as a male contraceptive agent due to its anti–steroidogenic activity but failed due to its high toxicity [107,108,109]. It was considered as an inhibitor for a variety of molecules including PKC1, LDHA or BCL–2 [110,111,112]. Inducing apoptosis by inhibiting BCL–2 [113,114], (−)–gossypol and its derivative AT–101 were evaluated in Phase II clinical trials for cancer treatment, including breast or prostate cancer [115,116]. In colon cancer cell lines, (−)–gossypol binds to MSI1 with higher affinity than to BCL–2 family members, repressing Notch/Wnt signaling in a MSI1–dependent manner [46]. Subsequently, (−)–gossypol was found to induce apoptosis and autophagy, reduce colon cancer cell proliferation and xenograft tumor growth in vivo [46].

Gossypolone, a major metabolite of gossypol, was identified as a potent inhibitor of MSI1, with more than 20–fold higher affinity than (−)–gossypol to disrupt MSI1 binding to the consensus motif [117]. Gossypolone was shown to inhibit both MSI1 and MSI2. Due to their low water solubility, PEGylated liposomes were introduced as a carrier to enhance delivery and thus potency in cellulo and in vivo [117]. Gossypolone–Liposomes exhibited a significant tumor inhibition efficacy and low systemic toxicity in mice. However, whether Gossypolone–directed cytotoxicity depends on MSI1 or BCL–2 inhibition, or both, needs further evaluation [118].

Luteolin was identified as the leading candidate of a 25,000–compound fluorescence polarization high–throughput screen (HTS), disrupting MSI1 binding to the consensus motif [88]. A direct interaction between MSI1 and Luteolin was confirmed by NMR [88]. The compound diminished MSI1′s positive impact on the expression of pro–oncogenic target genes and reduced proliferation, cell viability, colony formation, migration and invasion of glioblastoma cells. Luteolin, a common dietary flavonoid, possesses anti–oxidative, anti–inflammatory and anti–apoptotic activities in cancer and cardiovascular disease [119,120,121,122,123,124]. Due to its potent anti–tumor and anti–inflammatory effects, earlier studies and clinical trials on Luteolin had focused on cancer and inflammation (https://clinicaltrials.gov/ct2/show/NCT03288298, accessed on 25 March 2021) [125]. Luteolin treatment was found to inhibit epithelial–mesenchymal–transition (EMT) and promote apoptosis by repressing CREB1 and BCL–2 expression in colorectal cancer cells [126,127]. Xenograft and tail vain injection models of gastric and breast cancer cells revealed that Luteolin inhibits tumor growth, angiogenesis and metastasis by blocking Notch and VEGF signaling [128,129,130]. In glioblastoma, the compound was shown to prevent EGFR–mediated proliferation, promoting ROS–induced apoptosis [131].

MSI1 RNA–binding activity was also allosterically inhibited by ω−9 monounsaturated fatty acids (e.g., oleic acid) [132]. Oleic acid is a fatty acid found in olive oil and other plant oils. Moreover, it is the most abundant fatty acid in body fat, produced by mature oligodendrocytes during myelination [133]. Inducing MSI1 conformational changes that prevent MSI1–RNA association, oleic acid limits oligodendrocyte progenitor cell line proliferation [132]. However, it remains unclear if oleic acid inhibits protein–RNA interactions in general and which cellular mechanisms and pathways are modulated by this substance. However, several studies have reported an inhibition in cell proliferation induced by oleic acid in various tumor cell lines [134,135,136,137].

## 6. Conclusions

As an RNA–binding protein, MSI1 is a potent stemness factor with roles in tumor biology implicated by expression and function. Although its binding motifs are well defined in comparison to other RBPs, its mechanisms of action yet are not fully understood. As a post–transcriptional regulator of gene expression, MSI1 controls mRNA turnover and translation of target transcripts, modulating a variety of signaling pathways during development and cancer progression. Prospectively, MSI1 could be implicated in RNA modification as described for other RBPs, since it was shown to promote the expression of m6A readers such as YTHDF1. Whether MSI1 binding to RNA is sensitive to RNA modifications remains to be determined in the future. Besides its well–known roles in cell cycle regulation and Notch/Wnt signaling, MSI1 has lately been shown to be relevant for other pathways such as DNA damage response and repair. Although not highly specific at present, a number of novel MSI1–directed inhibitors already show promising anti–tumor potential. From this point of view, MSI1 solidifies its proposed role as a promising target candidate for cancer therapy, potentially as part of a combination therapy in a set of strategies, including differentiation therapy, Notch/Wnt antagonists and CDK or PARP inhibitors.

## Figures and Tables

**Figure 1 biology-10-00407-f001:**
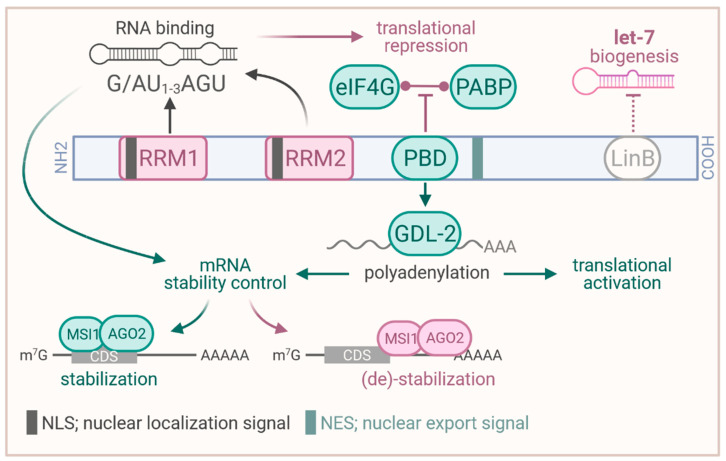
Structure of the MSI1 protein and proposed functions. RRM, RNA recognition motif; PBD; PABP binding domain (binding site to PABP and GLD–2); PABP, poly(A) binding protein; LinB, binding site to LIN28B; CDS, coding sequence. Created with www.BioRender.com (accessed on 29 April 2021).

**Figure 2 biology-10-00407-f002:**
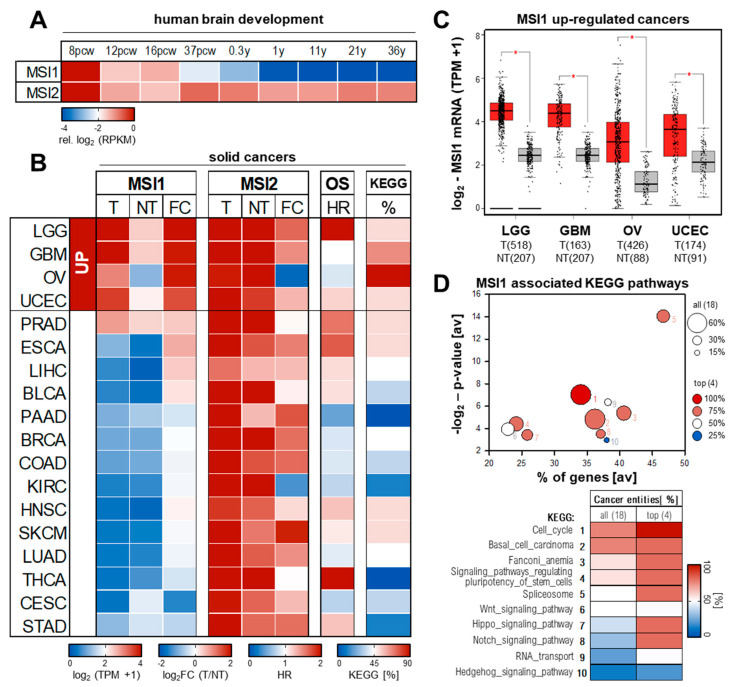
MSI1 expression and conserved association to hallmark of cancer pathways. (**A**) MSI1 and MSI2 expression in human brain development. Expression values (from https://www.brainspan.org/, accessed on 30 March 2021) in the neural plate determined by RNA–Seq is shown relative to 8 pcw. Pcw, post conception week; y, years. (**B**–**D**) Average (**B**) and total (**C**) expression values of MSI1 and MSI2 for tumor (T; TCGA, red) and non–tumor (NT; TCGA and GTEx, gray) tissues as well as fold changes (FC) of tumor vs. non–tumor tissue were obtained from the Gepia2 database (http://gepia2.cancer–pku.cn, accessed on 30 March 2021). Association of MSI1 with overall survival (B; OS) indicated as hazardous ratio (HR) was determined by KM plotter (https://kmplot.com/, accessed on 30 March 2021). The association of MSI1 positively correlated genes with cancer hallmark related KEGG (Kyoto Encyclopedia of Genes and Genomes) pathways was determined using the R2 database (https://hgserver1.amc.nl/, accessed on 30 March 2021). Top 10 of conserved associated pathways are shown as bubble diagram and heatmap (**D**) with number indicating ranking. The percentage of altered KEGG pathways (top 10) for the respective tumor entities is shown as heatmap (B, KEGG). BLCA, Bladder Urothelial Carcinoma; BRCA, Breast invasive carcinoma; CESC, Cervical squamous cell carcinoma and endocervical adenocarcinoma; COAD, Colon adenocarcinoma; ESCA, Esophageal carcinoma; GBM, Glioblastoma multiforme; HNSC, Head and Neck squamous cell carcinoma; KIRC, Kidney renal clear cell carcinoma; LGG, Brain Lower Grade Glioma; LIHC, Liver hepatocellular carcinoma; LUAD, Lung adenocarcinoma; OV, Ovarian serous cystadenocarcinoma; PAAD, Pancreatic adenocarcinoma; PRAD, Prostate adenocarcinoma; SKCM, Skin Cutaneous Melanoma; STAD, Stomach adenocarcinoma; THCA, Thyroid carcinoma; UCEC, Uterine Corpus Endometrial Carcinoma.

**Figure 3 biology-10-00407-f003:**
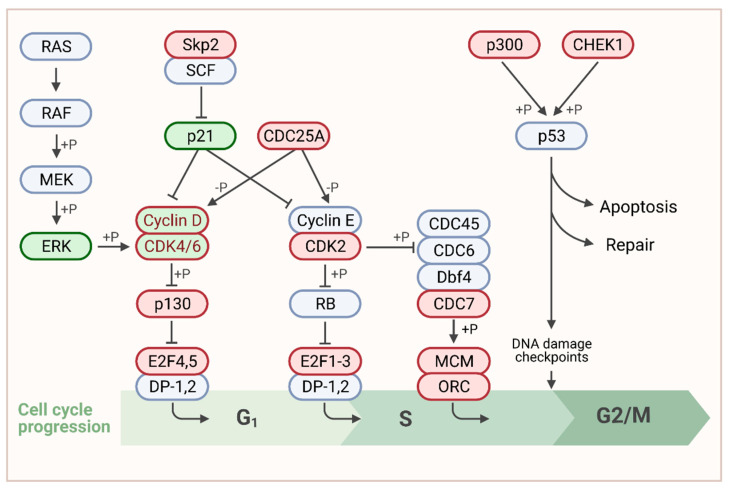
Association of MSI1 expression with cell cycle related genes. Schematic representation of the G1/S focus part of the KEGG: cell_cycle pathway according to (https://www.genome.jp/, accessed on 30 March 2021). For the entire pathway please refer to Appendix A. Published MSI1 target transcripts are depicted in green. Genes that were positively correlated to MSI1 in more than 50% of cancer entities are shown in red. Genes that belong to both categories are shown in dual color. Created with www.BioRender.com, accessed on 31 March 2021.

## Data Availability

Not applicable.

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
