# Peer review of "Musashi–1—A Stemness RBP for Cancer Therapy?"

_biology, 2021, doi:10.3390/biology10050407_

Round 1

Reviewer 1 Report

In the review by Bley and collaborators entitled “Musashi-1 - a stemness RBP for cancer therapy?” the authors summarize structure and functions of an RNA-binding protein Musashi-1 covering promotion of stemness during development in fly, flog and mammals. The authors then provide re-investigated results of the expression of Musashi-1 in solid human cancers and the associated genes with Musashi-1 using databases. Further, the authors focus on potential role of Musashi-1 as a stemness factor in brain, colon and female cancers and recent development of Musashi-1-directed inhibitors with anti-tumor activities. The manuscript is well written and touches on many aspects that can be of interest to a broad readership from scientists interested in RNA-binding proteins, stemness of cells and cancer therapies. However, it is somewhat confusing due to the large number of acronyms and abbreviations. A list of them would be helpful to the readers. I have only miner specific comments:

Many acronyms of proteins and methods, such as YTHDF1, PLF1, DKK3, SELEX and iCLIP, are not defined, please add full protein names and methods. A more description of proteins and methods as they are referenced multiple times in the manuscript would be helpful to understand how Musashi-1 functions as an RBP.

Lines 14 and 23: “Musashi1” replace by: Musashi-1 (with hyphen).

Line 15: “Oocyte” replace by: oocyte.

Figure 1: “NLS; nuclear localization signal” indicated by light gray box, not by gray box, replace by: “NES; nuclear export signal”. Clarify the difference between “PABP” protein and “PABP-binding domain” in Musashi-1. Suggest adding “CDS” (coding sequence) in the gray box of the mRNA.

Please keep affinity consistent through the text as follows:
Lines 106, 169, 311 and 361: “CDKN1A (p21/WAF)”, “CDKN1A (p21/WAF-1)”, “CDKN1A/p21” or “CDKN1A (p21)”
Lines 165 and 262: “Loss- and Gain-of-Function” or “loss- as well as gain-of-function” (uppercase or lowercase letters).

Line 107: Suggest adding (cyclin D1) to “CCND1”.

Line 110: Suggest adding (N6-methyladenosine) to “m6A”.

Line 124: “Namakura” replace by: Nakamura

Lines 139 and 157: “(GU3-6(G/AG)” and “(G/A)UnAGU)” replace by: GU3-6G/AG and G/AUnAGU, respectively.

Lines 169 and 341, Figure 3: “G2M” replace by: G2/M.

Abbreviations must be defined upon first use in the text.
Lines 226 and 303: “cancer stem cells (CSC)”
Lines 267 and 420: “epithelial-mesenchymal-transition (EMT)”

Figure 2C: please specify what the red and gray box-plots represent: tumor (T) and non-tumor (NT), respectively?

Line 239: “optained” replace by: obtained

Line 274: “Figure 2C” replace by: Figure 2D

Figure 3: “CD25” replace by: CDC25A

Author Response

  • We thank this referee for the kind review of our work.

Many acronyms of proteins and methods, such as YTHDF1, PLF1, DKK3, SELEX and iCLIP, are not defined, please add full protein names and methods. A more description of proteins and methods as they are referenced multiple times in the manuscript would be helpful to understand how Musashi-1 functions as an RBP.

  • We agree with this referee and added more descriptions throughout the manuscript (e.g. lines 105, 132, 138, 145, 185, 204, 239, 259, 358, including 2 footnotes for SELEX and iCLIP)

Lines 14 and 23: “Musashi1” replace by: Musashi-1 (with hyphen).

  • We corrected this

Line 15: “Oocyte” replace by: oocyte.

  • We corrected this

Figure 1: “NLS; nuclear localization signal” indicated by light gray box, not by gray box, replace by: “NES; nuclear export signal”. Clarify the difference between “PABP” protein and “PABP-binding domain” in Musashi-1. Suggest adding “CDS” (coding sequence) in the gray box of the mRNA.

  • We agree and corrected this by replacing the PABP referring to the PABP binding domain by PBD, added CDS to the coding sequence of the RNAs and corrected the light gray NLS to NES

Please keep affinity consistent through the text as follows:
Lines 106, 169, 311 and 361: “CDKN1A (p21/WAF)”, “CDKN1A (p21/WAF-1)”, “CDKN1A/p21” or “CDKN1A (p21)”

  • We constantly use now CDKN1A (p21/WAF)

Lines 165 and 262: “Loss- and Gain-of-Function” or “loss- as well as gain-of-function” (uppercase or lowercase letters).

  • We constantly stick to lowercase

Line 107: Suggest adding (cyclin D1) to “CCND1”.

  • We added this

Line 110: Suggest adding (N6-methyladenosine) to “m6A”.

  • We added this

Line 124: “Namakura” replace by: Nakamura

  • We corrected this

Lines 139 and 157: “(GU3-6(G/AG)” and “(G/A)UnAGU)” replace by: GU3-6G/AG and G/AUnAGU, respectively.

  • We replaced this as suggested

Lines 169 and 341, Figure 3: “G2M” replace by: G2/M.

  • We replaced this as suggested as well

Abbreviations must be defined upon first use in the text.
Lines 226 and 303: “cancer stem cells (CSC)”
Lines 267 and 420: “epithelial-mesenchymal-transition (EMT)”

  • We agree and added the definition

Figure 2C: please specify what the red and gray box-plots represent: tumor (T) and non-tumor (NT), respectively?

  • We added this in the figure legend

Line 239: “optained” replace by: obtained

  • We corrected this

Line 274: “Figure 2C” replace by: Figure 2D

  • We corrected this

Figure 3: “CD25” replace by: CDC25A

  • We thank this referee for all the suggestions. Listing the things that should be corrected was very helpful. We addressed everything in the revised version of the manuscript, corrected the 2 mistakes in Figure 1 and 3 and apologize for these.

Reviewer 2 Report

Title: Musashi-1 – a stemness RBP for cancer therapy? In this manuscript, the authors reviewed the role of RNA binding protein MSI1 in cancer therapy, particularly the review focused on MSI1 pathways, expression, and stemness factor. Further, the manuscript describes the recent advancement in identifying small molecule inhibitors for targeting the MSI1 – RNA interactions. Overall, this is a useful review study that aims to provide another way to understand the RNA binding protein MSI1 as a target for cancer therapy.

Author Response

We thank this referee for the kind review of our work.

Reviewer 3 Report

This manuscript is an excellent review of the history of the role of Musashi1, from the point of view of development, cancer, and its potential as a target for cancer therapy.  Also included in the MS is an analysis by the authors of publicly available data relating to MSH1 and 2 expression in solid tumors, survival rates, and association to KEGG pathways.  The paper is thorough and clear and should be of interest to a wide audience of scientists. 

A couple of minor questions:

  1. In figure 1, it is not clear what the difference is between the NLS signal marked by a dark rectangle and the NLS signal marked by a grey signal.
  2. p7 line 221 – should that be “striking” association?

Author Response

  • We thank this referee for the kind review of our work.

A couple of minor questions: 

  1. In figure 1, it is not clear what the difference is between the NLS signal marked by a dark rectangle and the NLS signal marked by a grey signal.
  • We apologize for this mistake and corrected this in the revised figure by replacing the light gray “NLS” to the correct label “NES”
  1. p7 line 221 – should that be “striking” association?
  • We apologize for this mistake and corrected this.